nanotechnology/physical chemistry/biochemistry

nano-scale metal-organic framework (NMOF), rifampicin, targeted drug delivery, antimicrobial action, anti-biofilm effect, methicillin-resistant *Staphylococcus aureus* (MRSA)

**Authors for correspondence:**
Saleh A. Ahmed
e-mail: saahmed@uqu.edu.sa
Samir Kumar Pal
e-mail: skpal@bose.res.in

# Nano-MOFs as targeted drug delivery agents to combat antibiotic-resistant bacterial infections

Saleh A. Ahmed[1,2], Md. Nur Hasan[3], Damayanti Bagchi[3], Hatem M. Altass[1], Moataz Morad[1], Ismail I. Althagafi[1], Ahmed M. Hameed[1], Ali Sayqal[1], Abd El Rahman S. Khder[1], Basim H. Asghar[1], Hanadi A. Katouah[1] and Samir Kumar Pal[3]

[1]Department of Chemistry, Faculty of Applied Science, Umm Al-Qura University, 21955 Makkah, Saudi Arabia
[2]Chemistry Department, Faculty of Science, Assiut University, 71516 Assiut, Egypt
[3]Department of Chemical, Biological and Macromolecular Sciences, S. N. Bose National Centre for Basic Sciences, Block JD, Sector III, SaltLake, Kolkata 700 106, India

SAA, 0000-0002-2364-0380; SKP, 0000-0001-6943-5828

The drug resistance of bacteria is a significant threat to human civilization while the action of antibiotics against drug-resistant bacteria is severely limited owing to the hydrophobic nature of drug molecules, which unquestionably inhibit its permanency for clinical applications. The antibacterial action of nanomaterials offers major modalities to combat drug resistance of bacteria. The current work reports the use of nano–metal-organic frameworks encapsulating drug molecules to enhance its antibacterial activity against model drug-resistant bacteria and biofilm of the bacteria. We have attached rifampicin (RF), a well-documented antituberculosis drug with tremendous pharmacological significance, into the pore surface of zeolitic imidazolate framework 8 (ZIF8) by a simple synthetic procedure. The synthesized ZIF8 has been characterized using the X-ray diffraction (XRD) method before and after drug encapsulation. The electron microscopic strategies such as scanning electron microscope and transmission electron microscope methods were performed to characterize the binding between ZIF8 and RF. We have also performed picosecond-resolved fluorescence spectroscopy to validate the formation of the ZIF8-RF nanohybrids (NHs). The drug release profile experiment demonstrates that ZIF8-RF depicts pH-responsive drug delivery and is ideal for targeting bacterial disease corresponding to its inherent acidic nature. Most

remarkably, ZIF8-RF gives enhanced antibacterial activity against methicillin-resistant *Staphylococcus aureus* bacteria and also prompts entire damage of structurally robust bacterial biofilms. Overall, the present study depicts a detailed physical insight for manufactured antibiotic-encapsulated NHs presenting tremendous antimicrobial activity that can be beneficial for manifold practical applications.

# 1. Introduction

Antibiotic resistance implies significant persistence of bacteria that are capable of growing in the presence of single or more antibiotics [1]. Multidrug resistance of bacteria is a significant danger to human civilization and it progresses at a faster speed owing to inappropriate and frequent use of antibiotics [2,3]. Lack of effective antibiotics will make general infections like bacterial pneumonia and compound procedures such as open-heart surgery much more dangerous and life-threatening [4,5] in future. Overuse of antibiotics, expanded time of treatment, uses of subtherapeutic dosage, prophylactic use of antibiotic agents and misuse of antibiotics for other non-bacterial diseases restricts its applicability followed by resistance generation [6,7]. Therefore, it is necessary to develop a new alternative strategy (targeted and stimuli responsive) to combat bacterial infections.

The indestructible superbugs represent a keen danger to worldwide wellbeing [8,9], demonstrating the beginning of the post-antibiotic period. In excess of 2 million patients have a serious ailment and over 20 000 passing because of irresistible illness from antibiotic safe microorganisms happen every year in the United States [10]. Ongoing reports foresee 10 million yearly deaths from bacterial contamination by 2050 [11]. Regardless of unbending cleanliness conventions pursued inside medical clinic conditions, most dangerous diseases are created in emergency clinics and human services related situations, which are normally named as healthcare-associated infections (HAIs) [12,13]. HAIs are frequently connected with Gram-positive microbes, for example, *Staphylococcus epidermidis* and *Staphylococcus aureus* infections [13]. The drug-resistant strain is positioned fifth in 11 as extremely dangerous bacteria to human civilization [11,14–16]. In this view, alternative antimicrobial treatment is one of the most encouraging and novel methodologies.

There is an appeal for improvement by employing stimuli-responsive nanohybrids (NHs) that can be accountable for its multifunction, multiuse and profoundly successful execution [17,18]. These days, nanoparticles (NPs) are progressively used to target microorganisms and could fill in as an option to natural antibacterial operators [19]. The viability of antibacterial capability might be expanded by means of a NH that could be powerful enough to prevent the development of microbes in humans [3]. The NH is probably going to keep stability against degradation, improving bioavailability and intracellular transportation [20,21].

Metal-organic frameworks (MOFs) are types of permeable materials containing metal sites attached through organic bridging ligands forming a frame-like structure and are appreciated broadly owing to large surface area, tunable hole arrangement and manageable functionalities [22,23]. Zeolitic imiadazolate framework 8 (ZIF8) is thermally nearly as steady as a covalent bond which makes it an extraordinary nominee for various applications [24]. Past reports have exhibited the use of ZIF8 as stimuli-responsive targeted delivery framework for typical anti-cancer treatment doxorubicin [25]. Under some conditions, ZIF8 crystals can course through the circulation system pursued by moderate discharge into intracellular organelles with lower (pH 5–6), portraying its likelihood to usage as pH-sensitive drug delivery [3,26]. We have shown earlier that ZIF8 can effectively deliver phototherapeutic drugs into a bacterial infection site and pronouncedly killed the infection using antimicrobial photodynamic therapeutic strategies [3]. Now, our attention is on improvement for increasing aqueous stability and the bioavailability issues of widely used antibiotics using ZIF8 [27].

In this work, we have used rifampicin (RF), a compelling antituberculosis drug that has tremendous pharmacological significance [28]. The common antimicrobial RF covers a wide scope of natural action while these investigations propose the use of ZIF8 frameworks for focused targeted drug delivery operators to combat antimicrobial resistance bacterial infections. RF is encapsulated into ZIF8 by a simple synthetic procedure [29]. The addition of RF inside the void available within the ZIF8 structure is required to bring down free dissemination of RF in biological media and point of confinement to the RF conglomeration potential outcomes. The synthesized ZIF8-RF NHs are characterized by electron microscopic methods as transmission electron microscope (TEM), scanning electron microscope (SEM) and diffraction strategy as X-ray diffraction (XRD). We have performed optical spectroscopic experiments such as absorption, emission and picosecond-resolved fluorescence spectroscopy to

validate the formation of the ZIF8-RF NH. The drug release profile experiment demonstrates that ZIF8-RF carries on as pH-responsive drug delivery and is ideal for targeting the bacterial disease. Most notably, ZIF8-RF provides enormous antibacterial activity against methicillin-resistant *Staphylococcus aureus* (MRSA) bacteria and prompts total loss of adherence to robust bacterial biofilms. The overall study depicts detailed physical insight for an antibiotic implanted ZIF8-RF hybrid with tremendous antimicrobial activity that can be beneficial for manifold biomedical application purposes.

# 2. Materials and methods

## 2.1. Chemicals

The chemicals $Zn(NO_3)_2,6H_2O$, 98% (zinc nitrate hexahydrate), rifampicin and 2-methyl imidazole (Hmim, 99%)) were purchased from Sigma. The reagents (methanol (99.8%, RANKEM) and ethanol (ACS reagent)) were used during the synthesis process. The DCFH-DA (2, 7-dichlorodihydrofluorescein diacetate) from Calbiochem and dimethyl sulfoxide (DMSO) from Merck were used. Unfiltered $H_2O$ from Millipore was used as aqueous solvent.

## 2.2. Fabrication of zeolitic imiadazolate framework 8 (ZIF8) and ZIF8-rifampicin

ZIF8 NPs were produced following the previously described sol–gel method [26]. In short, 3.45 mg of $Zn(NO_3)_2,6H_2O$ and 81 mg of 2-methyl imidazole were solubilized individually in 5 ml methanol solution. After dissolution, the 2-methyl imidazole was added to zinc nitrate hexahydrate under the constant stirring condition and then kept in a stable condition for 24 h. The synthesized samples were washed with methanol solvent by centrifugation and dried at normal room temperature. Ten milligrams of synthesized ZIF8 sample was added to 0.5 mM of RF solution in DMSO solvent under a constant stirring condition and the whole solution was allowed to stir for a night. Finally, the ZIF8-RF sample was collected by washing with DMSO and then with distilled water and dried at 90°C. The RF drug loading capability (DLC) was measured in an ethanol solution of RF. The absorption peak of RF was collected before and after loading onto the ZIF8 sample and by relating the absorption peak intensity at 480 nm, DLC (%) was calculated as per the below equation:

$$\frac{\text{the amount of drug loaded}}{\text{the amount of ZIF8 sample}} \times 100.$$

## 2.3. Characterization methods

The crystal nature of the integrated ZIF8 was found by using powder XRD as shown in our previous reports [23]. The electron microscopic investigation such as field emission SEM (FESEM, QUANTAFEG 250) and TEM (Tecnai S-Twin, working at 200 kV) was performed for both the samples (ZIF8 and ZIF8-RF) by dispersing in ethanol media. The thermogravimetric analysis (TGA) experiment for synthesized samples was implemented in nitrogen media from 40°C to 650°C temperature at a pace of 10°C $min^{-1}$ by TGA-50H (PerkinElmer). A Shimadzu spectrophotometer (UV-2600) was used for the measurement of absorbance spectra of the samples in aqueous media. Fluorolog (HORIBA) was used to measure the steady state emission spectra and the Edinburgh time-resolved instrument was used to perform time-correlated single photon counting decay as reported in our earlier reports [30,31].

## 2.4. Details of bacterial culture methods used

Bacterial culture analysis was evaluated using MRSA (strain 1692) as the typical organism. The MRSA bacteria growth in Luria Broth (LB) media was performed in a shaker incubator at 37°C. Every one of the experiments was finished with naturally developed overnight culture. The colony-forming unit (CFU) assay was performed on MRSA bacterial cells. Once the optical density (OD) reached approximately 0.6 in the absorbance band, the MRSA bacterial culture was diluted $10^3$ times. The culture was treated by ZIF8-RF (for four different concentrations C1: 0.25 mg $ml^{-1}$ sample, C2: 0.50 mg $ml^{-1}$ sample, C3: 0.75 mg $ml^{-1}$ sample and C4: 1.00 mg $ml^{-1}$ sample of ZIF8-RF) and RF (maintaining the OD at 480 nm similar to the ZIF8-RF NH) for 3 h. A hundred microliters from each of the treated samples were spread on agar plates and incubated at 37°C overnight. The CFU were counted after incubation and triplicates of the experiment were carried out, and the results were expressed with a statistical *p*-value less than or equal to 0.05.

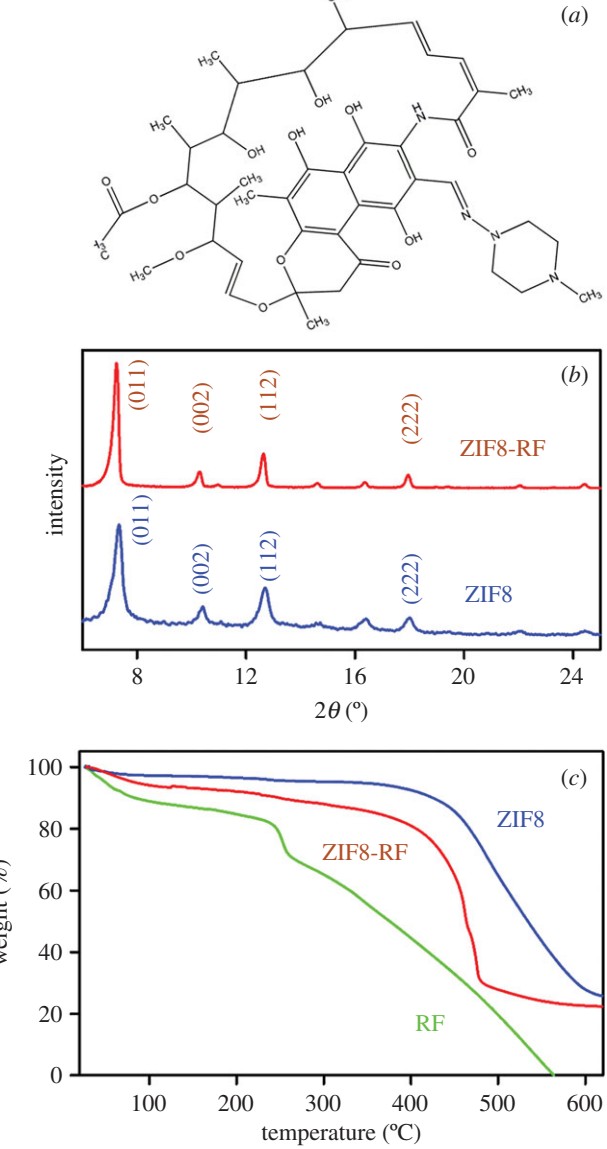

**Figure 1.** (*a*) Structure of rifampicin (*b*) XRD patterns of ZIF8 (blue) and ZIF8-RF (red). (*c*) Thermogravimetric profile of ZIF8 (blue), RF (green) and ZIF8-RF (red) monitored under $N_2$ flow.

## 2.5. Bacterial biofilm development technique

The biofilms of MRSA bacteria were prepared in LB media on 60 mm antiseptic polycarbonate Petri plates. These cells were uniformly spread over the Petri plate and then incubated for 2 days at 37°C [18]. Crystal violet (CV; 0.1% (w/v)) assay analysis was used to measure the biofilms. Then it was solubilized in 95% ethanol. The level of biofilm arrangement was assessed using a UV-visible (UV-Vis) absorbance band at 588 nm. The biofilms have also been cultured by the ZIF8-RF sample (concentration C3) with respect to control MRSA, and the morphological variations of the biofilms were obtained using SEM.

## 3. Results and discussion

The chemical structure of RF is depicted in figure 1*a*. Figure 1*b* represents the XRD peaks of synthesized ZIF8 and ZIF8-RF. XRD profiles of both ZIF8 and ZIF8-RF show XRD peaks at $2\theta = 7.2°$, 10.3°, 12.5° and 17.8° with corresponding diffraction planes (011), (002), (112) and (222). The diffraction peaks are in good agreement with reported article [3] affirming that the synthesized sample has an unadulterated ZIF8 stage before and after the drug encapsulation. In any case, the XRD shape of ZIF8-RF (figure 1*b*)

displays comparative peaks to ZIF8 with a minor fluctuation as a tiny widening at lower values of 2θ in ZIF8-RF contrasted with those of ZIF8. The perception shows a reduction in the precious crystal size as indicated by Debye–Scherrer's equation [32]. The RF drug encapsulation into the micropores of ZIF8 is answerable for the adjustment in the interplanar distances.

The thermal stability of the samples is evaluated using TGA; the subsequent bends are shown in figure 1c. The primary weight reduction is around 10% before 450°C owing to the removal of solvent and some guest molecules. A significant difference in around 60% weight reduction after 500°C suggests the separation of the natural ligand which accounts for the collapse of the system structure. Along these, ZIF8 is thermally steady up to nearly 450°C. Whereas, in the thermal steadiness of RF, the underlying weight reduction is about 13% and happens at around 80°C, which may resemble the expulsion of the dissolvable particles. A significant difference of around 40% weight reduction after 250°C and complete weight reduction happens at around 500°C as appears in figure 1c. In the thermal stability existence of ZIF8-RF, the underlying weight reduction is about 10% and happens at the scope of 300°C, owing to the removal of the dissolvable particles. Moreover, ZIF8-RF starts thermal breakdown at 400°C, and almost 76% weight reduction happens within 500°C. The commencement of thermal breakdown at a prior temperature is ascribed to the separation of RF particles embodied inside the ZIF8 structure [33].

The morphological investigation of the synthesized NPs of ZIF8 and ZIF8-RF was confirmed using SEM, TEM and energy filtered TEM (EFTEM). The SEM picture of ZIF8 (figure 2a) represents an unchanging rhombic dodecahedron crystal measurement. The SEM pictures of ZIF8-RF (figure 2b) outline adjustment in edge thickness contrasted with ZIF8 with shrinkage of crystal size. No clear accumulation or damage in crystal structure was acquired by SEM that delineates that RF molecules are inserted into the pores inside or to the outside surface of the synthesized ZIF8 system [34]. Figure 2c represents the particle size distribution of ZIF8 (average particle size 158.72 ± 0.52 nm) and figure 2d represents the particle size distribution of ZIF8-RF (average particle size 157.96 ± 1.07 nm). The size of ZIF8 was unchanged before and after drug loading. Figure 2e depicted the schematic representation of the formation of ZIF8-RF NH. The TEM was performed for structural clarification of ZIF8 and ZIF8-RF. TEM results uncovered the size and morphology of ZIF8 (figure 3a) as a uniform rhombic dodecahedral shaped precious crystal with a normal distance across of 150–250 nm. To watch the synthetic structure of ZIF8 frameworks, we have used EFTEM mapping the experiment of ZIF8 (figure 3b). The EFTEM guide of multiple crystals of ZIF8 represents the spreading of metal (Zn) ions and ligand (N) atoms throughout the crystal. ZIF8-RF nano-conjugates are also described using TEM. Figure 3c represents the different precious crystal structures of the ZIF8-RF. The TEM picture portrays a slight reduction in normal measurement diameter. The EFTEM map (figure 3d) shows the spreading of metal (Zn) ions and ligand (N) atoms all through the precious ZIF8-RF. The perception proposes no evident collection or destruction in overall crystal structure was acquired, depicting that RF atoms are entangled into the pores of the ZIF8 system [34].

To investigate the optical properties of the frameworks, UV-Vis absorption bands were measured for every sample. Figure 4a shows the UV-Vis absorption of RF (green), ZIF8 (red) and ZIF8-RF (blue) in DMSO. The characteristic absorbance peaks of RF are at 330 nm and 480 nm, respectively. After conjugation, the trademark peaks of RF were seen in ZIF8-RF as appears in figure 4a (blue) which shows the effective complexation among RF and ZIF8 NPs. The room temperature photoluminescence spectra of RF (green) and ZIF8-RF (blue) appear in figure 4b, and the relating room temperature excitation spectra are given in the inset. It is seen that both RF and ZIF8-RF showed an extreme emission peak at 425 nm upon excitation at 375 nm. The excitation spectra are represented in the inset of figure 4b. In both cases, after the connection of RF to the ZIF8 NPs, the steady state emission intensity is essentially decreased which may show the atomic level interaction during the synthesis process. We have employed Förster resonance energy transfer (FRET) from the ZIF8 surface to the nearby RF because of their significant spectral overlap as appear in figure 4c. The steady state emission intensity of ZIF8 in the NH quenched sufficiently compared to that of free ZIF8, which can be ascribed to the effective nonradiative photo-induced procedures from ZIF8 to the RF [35]. Thus, we propose FRET from the donor of ZIF8 NPs to RF, the acceptor [36]. The assessment of atomic separations in various gatherings from the FRET calculation has become a useful tool [37,38]. The fluorescence decay of the ZIF8 in the presence and absence of the acceptor RF was acquired upon excitation with a 375 nm laser and collected at 450 nm (figure 4d). The detailed information of the fitting specification of the fluorescence decay is presented in table 1. From FRET formulations, the separation between the donor ZIF8 NPs and acceptor RF is found to be 13.3 Å. The energy transfer efficiency is determined to be 65%. The FRET separations confirm the vicinity of the RF to the ZIF8 [36].

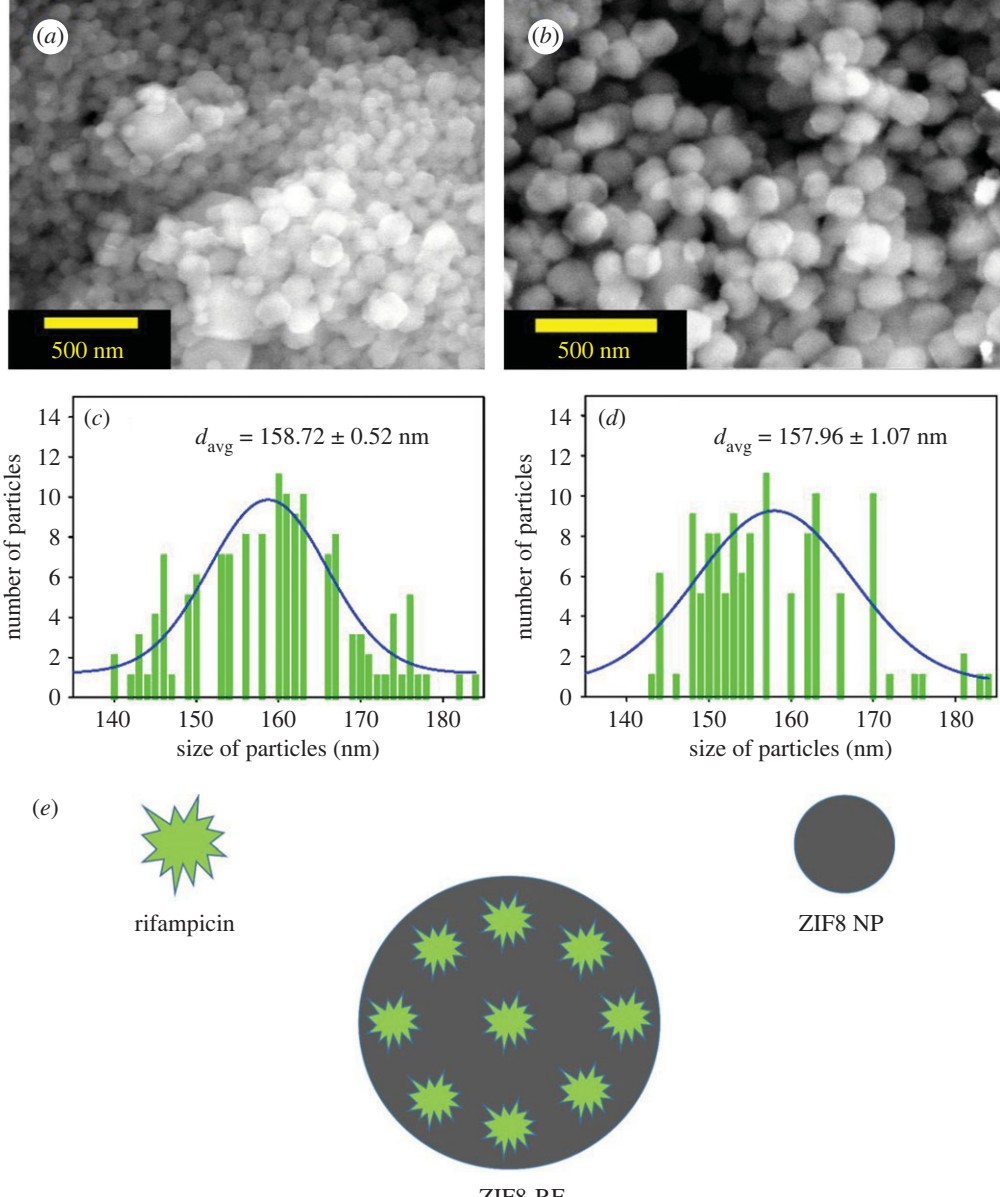

**Figure 2.** (*a*) The FESEM image of ZIF8. (*b*) FESEM image of ZIF8-RF. (*b*) Particle distributions of ZIF8 (average particle size 158.72 ± 0.52 nm). (*d*) Particle distributions of ZIF8-RF (average particle size 157.96 ± 1.07 nm). (*e*) Schematic representation of ZIF8-RF NH.

From that point, the time-dependent drug release profile experiment of ZIF8-RF performed at two distinct pH conditions. The drug release profile of ZIF8-RF measured up to 8 h for the time interval of 2 h as appears in figure 5*a*. We have found significant RF release (69%) with respect to control ZIF8. It has to be noted that if a drug is not been released for a longer period of time (between 5 and 12 h after the administration of single dose) it may be excreted from our body [39]. The higher metabolic degree owing to anaerobic fermentation, microbial diseases regularly have acidic situations [40]; ZIF8-RF is anticipated to be invested more in the diseased site than ordinary tissues. These outcomes demonstrate that ZIF8-RF continues as a pH-responsive treatment, as the Zn–N coordination in ZIF8 will, in general, dissociate at pH of approximately 5.0. The Zn–N dissociation makes the pH-responsive drug release, and owing to this reason, ZIF8 is ideal for targeting the bacterial disease [3,41]. Scheme 1 depicts the schematic representation of nano-MOFs as targeted drug delivery agents to combat antibiotic-resistant bacterial infections. The RF DLC of ZIF8 was calculated by measuring the absorption peak of RF at approximately 480 nm and DLC of ZIF-8 was 42%. Next, we surveyed the antimicrobial activity of ZIF8-RF towards MRSA bacteria. The pH of LB culture medium reduces with bacterial growth as the cell density increases [42]. We have performed control experiments using

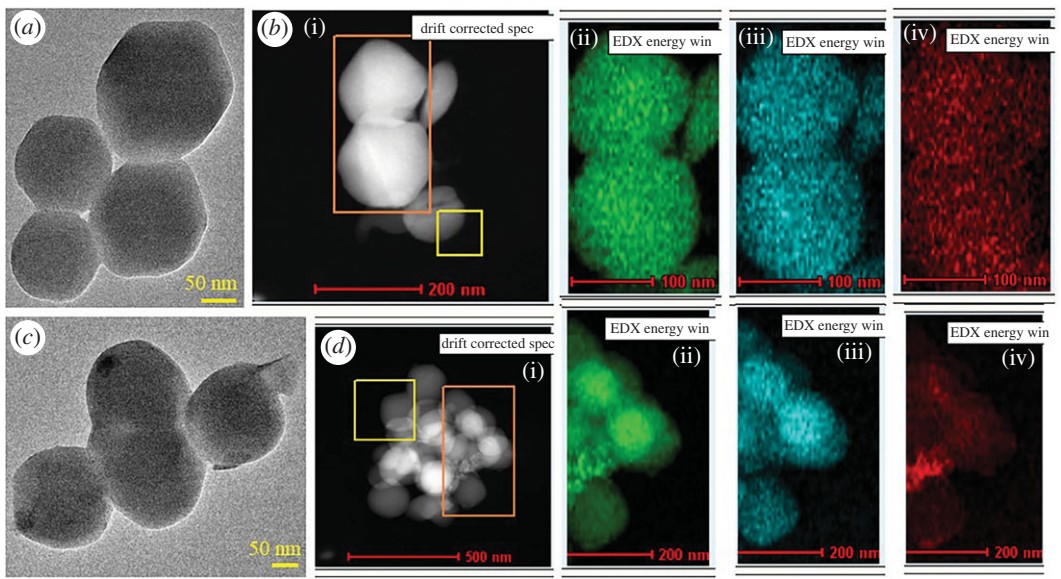

**Figure 3.** (*a*) TEM image of ZIF8. (*b*) EDAX mapping of ZIF8. (*c*) TEM image of ZIF8-RF. (*d*) EDAX mapping of ZIF8-RF. Green dots signify Zn, blue dots signify N and red dots denote O.

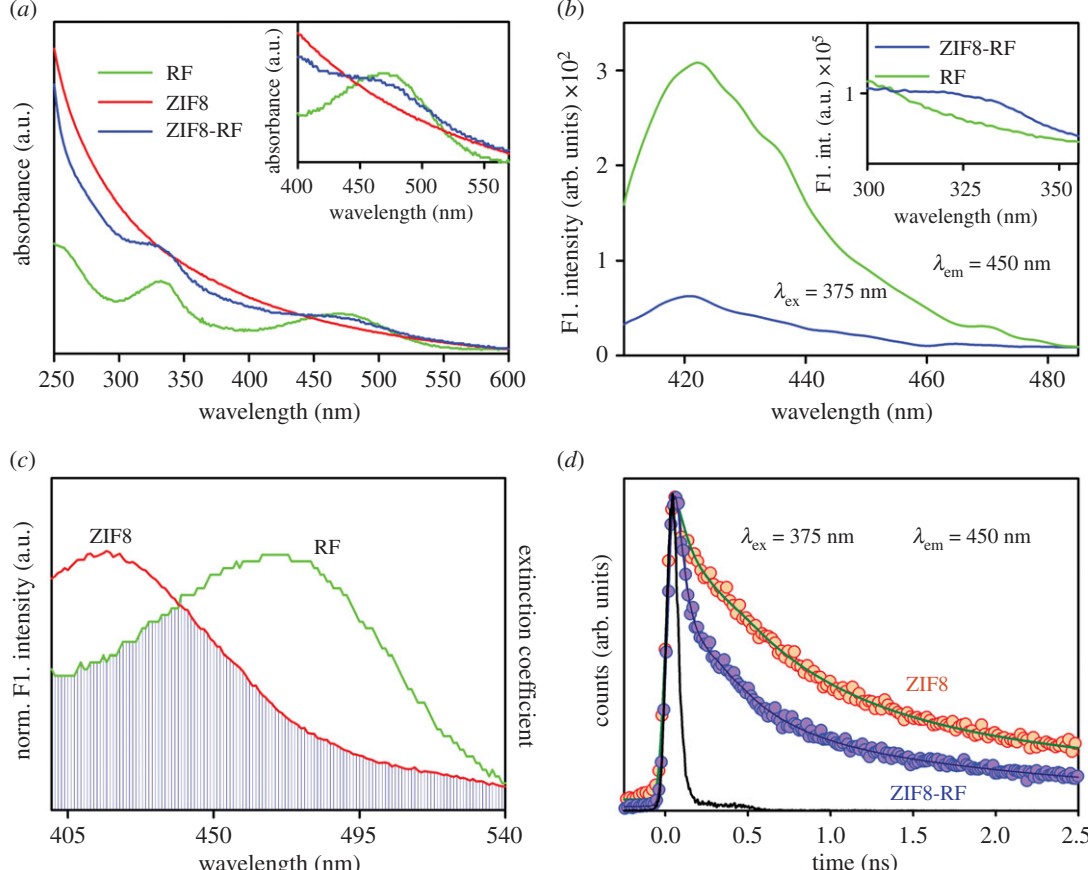

**Figure 4.** (*a*) UV-Visible absorbance of ZIF8-RF (blue), RF (green) and ZIF8 (red). (*b*) Room temperature photoluminescence spectra (excitation at 375 nm) of RF (green) and ZIF8-RF (blue). Inset shows steady state excitation spectra of RF (green) and ZIF8-RF (blue) at 450 nm. (*c*) Spectral overlap of ZIF8 (donor) and RF (acceptor). (*d*) The picosecond-resolved fluorescence transient of ZIF8-RF (blue), RF (green) and ZIF8 (red). The excitation wavelength was 375 nm and the collection wavelength was 450 nm.

the same amount of ZIF8 to that of ZIF8-RF with 3 h incubation time. ZIF8 has not provided any significant change in the number of bacterial colonies compared to the untreated MRSA. This effect suggests that the delivery of RF plays a significant role in the antibacterial effect of ZIF8-RF

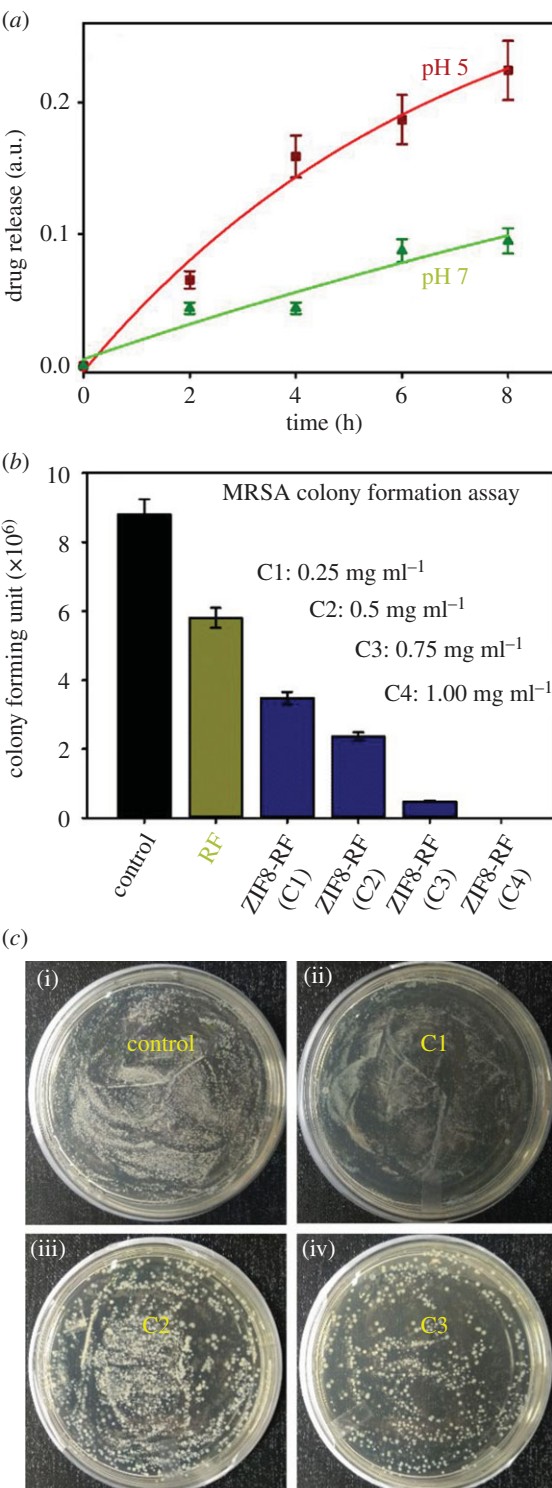

**Figure 5.** (*a*) Drug release nature of ZIF8-RF at two different pH conditions. (*b*) Dose-dependent antibacterial effect of ZIF8-RF at varying concentrations on methicillin-resistant *S. aureus*. (*c*) Images of *S. aureus*; agar plates treated with (i) no samples and (ii–iv) different concentrations of ZIF8-RF samples.

composite. Thus, we anticipated the lower pH condition due to bacterial growth, promote the loosening of bonds in porous ZIF8 and consequently improve the delivery of RF. The experiments for variable concentrations of ZIF8-RF were analysed as appear in figure 5*b*. For RF (green)-treated samples, an much less bacterial executing impact is seen (figure 5*b*). Figure 5*c* show MRSA plates treated with four different concentrations (C1: 0.25 mg ml$^{-1}$ sample, C2: 0.50 mg ml$^{-1}$ sample, C3: 0.75 mg ml$^{-1}$ sample and C4: 1.00 mg ml$^{-1}$) of ZIF8-RF. The visual distinction in the number of MRSA settlements

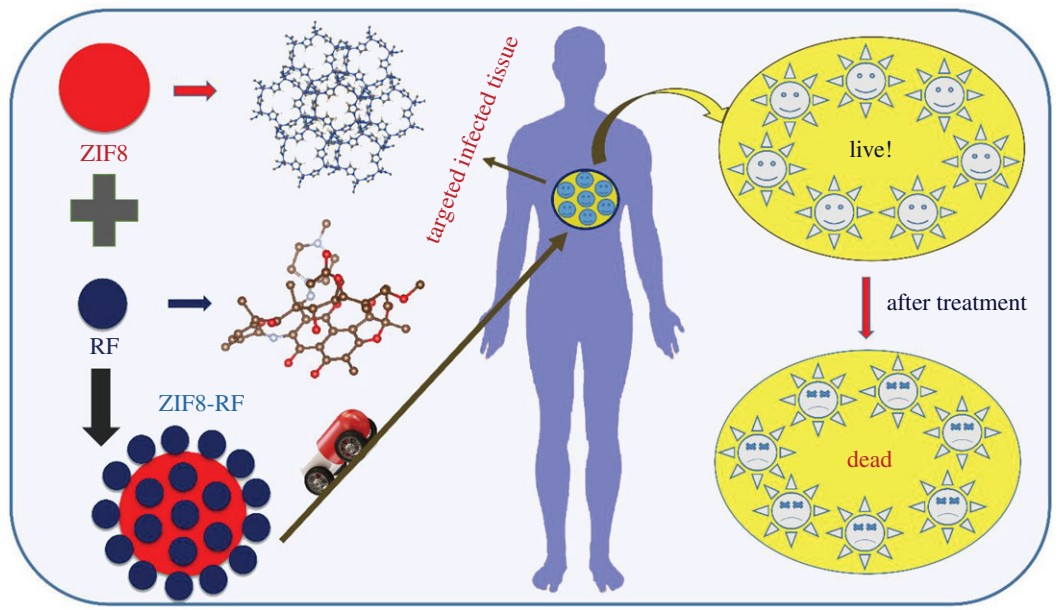

**Scheme 1.** Schematic representation of nano-MOFs as targeted drug delivery agents to combat antibiotic-resistant bacterial infections.

**Table 1.** Picosecond-resolved fluorescence transient lifetime. (The emission (monitored at 450 nm) was detected with 375 nm laser excitation. Numbers in parentheses indicate relative contributions.)

| system | $\tau_1$ (ps) | $\tau_2$ (ps) | $\tau_3$ (ps) | $\tau_{avg}$ (ps) |
|---|---|---|---|---|
| ZIF8 | 50 (40%) | 608 (40%) | 3510 (20%) | 965.2 |
| ZIF8-RF | 34 (69%) | 320 (20%) | 2253 (11%) | 335.3 |

after treating with ZIF8-RF nano-conjugate, antibacterial impact was approximately 96% colony reduction for C3 concentration and the minimum inhibitory concentration was found to be 1 mg ml$^{-1}$ (C4) of ZIF8-RF. The extraordinary antimicrobial viability of ZIF8-RF inspires us to examine the biofilm interruption ability of ZIF8-RF. Biofilms are infectious arrangements of free-living microbes connected to a surface or with one another using an extra polymeric substance, a common organic drug resistant [43]. To build up that MRSA 1692 can form biofilms, we first accomplished a crystal violet (CV) assay on overnight grown bacteria. The CV stained material that had been stored on the sides of the biofilm by the developing microbes, compared with CV staining of biofilm that contained just culture medium. Estimation of the OD at 588 nm of solubilized CV stain assisted this perception to be communicated in relative units (figure 6a). The intrinsic acidic condition of biofilms encourages us to do further annihilation activity by ZIF8-RF on MRSA bacteria. The morphological changes after treatment of the biofilms are exhibited by SEM pictures. Figure 6b displays the structure of the MRSA biofilms, and figure 6c represents treatment with ZIF8-RF. The verifiable colonization impacts are essentially diminished upon ZIF8-RF treatment. These perceptions affirm that ZIF8-RF is exceptionally successful against bacterial infections. The antibacterial effect by ZIF8-RF using the bioanalytical techniques condition is estimated to deliver momentous real-life applicability of the fabricated nano-conjugates.

# 4. Conclusion

In short, we present a simple synthetic encapsulation approach for the blend of a Zn-based organization polymeric framework (ZIF8) implanting a drug, RF. The synthesized NH with the drug particles entangled inside the well-characterized porous enclosure structure of ZIF8 prevents the self-aggregation of RF. ZIF8-RF nano-binding shows FRET from ZIF8 to RF that confirm the proximity of the RF to the ZIF8 with molecular resolution. The immense impact of ZIF8-RF toward MRSA even at

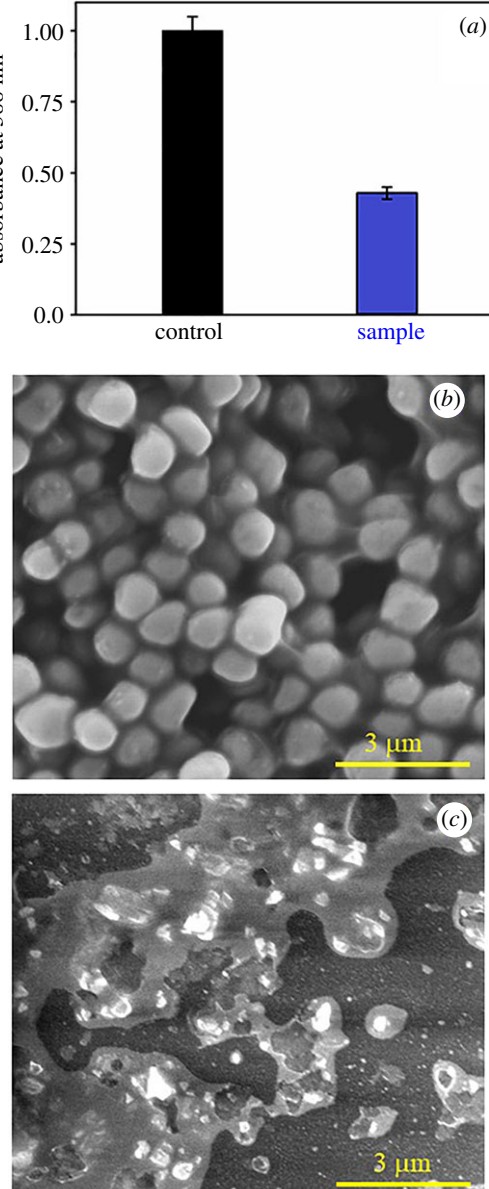

**Figure 6.** (*a*) Antibiofilm effects of ZIF8-RF on the MRSA biofilm monitored by the adhesion efficiency of biomass through the crystal violet staining assay. Field emission gun SEM images of (*b*) MRSA biofilms and (*c*) MRSA biofilms after ZIF-RF-treated samples. Scale bar is 3 μm.

particularly lower concentrations proposes its uniqueness and possible outcomes for the real applications. Generally, our present study exhibits significant efficacy in antibacterial effect through encapsulation in a biocompatible helpful nano-platform for the efficient remediation of drug resistance bacterial diseases.

Data accessibility. All the necessary data are included in the main manuscript; there is no supplementary data to declare and raw data for all the figures in the paper have been uploaded to the Dryad Digital Repository: https://doi.org/10.5061/dryad.31zcrjdhk [44].

Authors' contributions. S.A.A. has provided the research plan, assisted in scientific discussion, performing the experiments, analysed the data and contributed to writing the manuscript. M.N.H. has performed the experiments, prepared the figures and contributed to writing the manuscript. D.B. helped in performing experiments and assisted in manuscript writing. H.M.A. and M.M. helped in the preparation of figures and assisted in the manuscript preparation. I.I.A. and A.M.H. analysed the data and assisted in scientific discussion. A.S. and A.S.K. assisted in scientific discussion and assisted in manuscript writing. B.H.A. and H.A.K. has contributed to writing the manuscript. S.K.P. provided the research plan, assisted in scientific discussion and wrote the manuscript. All authors reviewed the manuscript.

Competing interests. There are no conflicts to declare.

Funding. This work was full budgetary supported by the grant code: 20UQU0041DSR.

Acknowledgement. The authors would like to thank the Deanship of Scientific Research at Umm Al-Qura University for fully supporting this work by the grant code: 20UQU0041DSR. M.N.H. thanks the CSIR, India for providing the fellowship. S.K.P. thanks DST, India for Abdul Kalam Technology Innovation National Fellowship (INAE/121/AKF).

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
