## [Reviewer comments · Royal Society Open Science]

Review History

RSOS-200959.R0 (Original submission)

Review form: Reviewer 1

Is the manuscript scientifically sound in its present form?

Yes

Are the interpretations and conclusions justified by the results?

Yes

Is the language acceptable?

Yes

Do you have any ethical concerns with this paper?

No

Have you any concerns about statistical analyses in this paper?

No

Recommendation?

Accept as is

Comments to the Author(s)

Professor Saleh and his coworkers report work in which they attach rifampicin (RF), a well-recognized antituberculosis drug, into the pore surface of zeolitic imidazolate framework 8 (ZIF8) by a simple and convenient synthetic procedure. The authors used X-ray diffraction (XRD) method to characterize the synthesized ZIF8 before and after drug encapsulation. In addition, they utilized scanning electron microscope (SEM) and transmission electron microscope (TEM) methods to characterize the binding between ZIF8 and RF. The authors have found that ZIF8-RF depicts pH-responsive drug delivery and is ideal for targeting bacterial disease due to its inherent acidic nature. One key finding, ZIF8-RF has been observed to display enhanced antibacterial activity against methicillin-resistant *S. aureus* (MRSA) bacteria and also inflicts full damage on structurally robust bacterial biofilms. The present study offers extensive physical insight for the manufactured antibiotic-encapsulated nanohybrids which are promising prospects and may prove beneficial for diverse practical applications.

General comments: I appreciate Scheme 1: Schematic representation of nano MOFs as targeted drug delivery agents to combat antibiotic-resistant bacterial infections. It is very informative. The research carried out by Dr. Saleh et al. will appeal to wide audience in areas such as medicinal chemistry, organic chemistry, drug design and drug delivery. The research is original and has not been published elsewhere. The present work contributes to the advancement of scientific knowledge. All conclusion drawn by the authors are supported by the data and data supporting the findings of the paper have been included in the manuscript. I believe that all experimental protocols/procedure and statistical analysis performed to a high technical standard which are both methodologically and scientifically sound. Characterisation of new compounds also meets the criteria outlined by the journal.

The manuscript is well suited for The Royal Society Open Science.

Kindly correct the following:

Corrections:

1. Add spaces between text and references throughout the manuscript. For example, wellbeing[8] should be corrected to wellbeing [8].
2. Change "We have also earlier showed that" to "We have showed earlier"
3. Remove spaces in the formula ($Zn(NO_3)_2 \cdot 6H_2O$)
4. Correct "ZIF8 NPs were produced succeeding in the recently revealed sol-gel method.25" to "ZIF8 NPs were produced following the recently described sol-gel method [25]."
5. Correct "A serious difference in around 60% weight reduction after 500 °C recommends the separation of the natural ligand which accounts for collapse of the system structure." To "A serious difference in around 60% weight reduction after 500 °C suggests the separation of the natural ligand which accounts for the collapse of the system structure."
6. Correct "To investigate the optical properties of the frameworks" to "To discover the optical properties of the frameworks,"
7. Correct "To discover the optical properties of the frameworks, UV-Vis absorption bands were estimated for every sample." To "To investigate the optical properties of the frameworks, UV-Vis absorption bands were measured for every sample."
8. Correct "as appeared in Figure 4a (Blue)" to "as appears in Figure 4a (Blue)"
9. Correct "become a useful tool.[36, 37] The" to "become a useful tool [36, 37]. The"

Review form: Reviewer 2

Is the manuscript scientifically sound in its present form?

No

Are the interpretations and conclusions justified by the results?

No

Is the language acceptable?

Yes

Do you have any ethical concerns with this paper?

No

Have you any concerns about statistical analyses in this paper?

No

Recommendation?

Major revision is needed (please make suggestions in comments)

Comments to the Author(s)

The article entitled "Nano MOFs as targeted drug delivery agents to combat antibiotic resistant bacterial infections." has provided novel nanomaterials for the treatment of drug resistant bacteria. The authors suggest that the utilization of ZIF8 to load antibiotics RF can help with bacteria target delivery of antibiotics with enhanced antibacterial efficacy. Overall, this is an interesting study and it is within the journal's scope, however the evidence are not so sufficient and there are some serious issues to be addressed.

1. The authors claimed that the size of ZIF8 was unchanged before and after drug loading, however in the size of nanoparticles of ZIF8 and ZIF8-RF seemed much different shown in Figure 2 a and b, please explain.
2. The releasing profile of 8 h can only reach 0.2 even at pH 5.0, the authors should provide the drug releasing profile for longer periods.
3. Figure 3 please use the same magnification of ZIF8 and ZIF8-RF.
4. RF is not a common antibiotic used in clinic for treating MRSA, why not using regular antibiotics such as ampicillin? Moreover, the authors should test more than one drug to support the idea of this nano-MOF as target drug carrier.
5. What's the drug loading rate of ZIF8-RF? What's the concentration of RF used as control? What's the reason for the enhanced antibacterial efficacy of ZIF8-RF towards bacterial than pure RF? The authors should also provide the antibacterial effect of ZIF8 as control group and explain the enhanced antibacterial efficacy in detail.
6. The authors claimed that the nano-MOF can be used to combat drug resistant bacteria, however the antibacterial mechanism of this new ZIF8-RF seems to rely on RF, which is also an antibiotic. And ZIF8 only serves as a drug delivery system, how can this system overcome drug resistance?
7. What's the pH of the culture medium at the end of the treatment? As we know that the pH of LB culture medium is 7.4, it is difficult to believe that 3 h treatment in it this environment can trigger the release of RF in ZIF8-RF.
8. There is still much bacteria after the treatment of C3, as we know that longterm release of antibiotics at low concentrations can stimulate the emergence of drug-resistant bacteria, the authors should discuss this in the manuscript and provide the MIC for the ZIF8-RF.

9. In Figure 6 for the assay of biofilm, the authors didn't provide the anti-biofilm effect of ZIF8 and RF as control.
10. Spelling mistake, such as Page 4 line 10, "dangerto".

Decision letter (RSOS-200959.R0)

Dear Professor Pal:

Title: Nano MOFs as targeted drug delivery agents to combat antibiotic resistant bacterial infections

Manuscript ID: RSOS-200959

The editor assigned to your manuscript has now received comments from reviewers. We would like you to revise your paper in accordance with the referee and Subject Editor suggestions which can be found below (not including confidential reports to the Editor). Please note this decision does not guarantee eventual acceptance.

Please submit your revised paper before 25-Jul-2020. Please note that the revision deadline will expire at 00.00am on this date. If we do not hear from you within this time then it will be assumed that the paper has been withdrawn. In exceptional circumstances, extensions may be possible if agreed with the Editorial Office in advance. We do not allow multiple rounds of revision so we urge you to make every effort to fully address all of the comments at this stage. If deemed necessary by the Editors, your manuscript will be sent back to one or more of the original reviewers for assessment. If the original reviewers are not available we may invite new reviewers.

RSC Associate Editor:
 Comments to the Author:
 (There are no comments.)

RSC Subject Editor:
 Comments to the Author:
 (There are no comments.)

Reviewers' Comments to Author:
 Reviewer: 1

Comments to the Author(s)

Professor Saleh and his coworkers report work in which they attach rifampicin (RF), a well-recognized antituberculosis drug, into the pore surface of zeolitic imidazolate framework 8 (ZIF8) by a simple and convenient synthetic procedure. The authors used X-ray diffraction (XRD) method to characterize the synthesized ZIF8 before and after drug encapsulation. In addition, they utilized scanning electron microscope (SEM) and transmission electron microscope (TEM) methods to characterize the binding between ZIF8 and RF. The authors have found that ZIF8-RF depicts pH-responsive drug delivery and is ideal for targeting bacterial disease due to its inherent acidic nature. One key finding, ZIF8-RF has been observed to display enhanced antibacterial activity against methicillin-resistant *S. aureus* (MRSA) bacteria and also inflicts full damage on structurally robust bacterial biofilms. The present study offers extensive physical insight for the manufactured antibiotic-encapsulated nanohybrids which are promising prospects and may prove beneficial for diverse practical applications.

General comments: I appreciate Scheme 1: Schematic representation of nano MOFs as targeted drug delivery agents to combat antibiotic-resistant bacterial infections. It is very informative. The research carried out by Dr. Saleh et al. will appeal to wide audience in areas such as medicinal chemistry, organic chemistry, drug design and drug delivery. The research is original and has not been published elsewhere. The present work contributes to the advancement of scientific knowledge. All conclusion drawn by the authors are supported by the data and data supporting the findings of the paper have been included in the manuscript. I believe that all experimental protocols/procedure and statistical analysis performed to a high technical standard which are both methodologically and scientifically sound. Characterisation of new compounds also meets the criteria outlined by the journal.

The manuscript is well suited for The Royal Society Open Science.

Kindly correct the following:

Corrections:

1. Add spaces between text and references throughout the manuscript. For example, wellbeing[8] should be corrected to wellbeing [8].
2. Change "We have also earlier showed that" to "We have showed earlier"
3. Remove spaces in the formula ($Zn(NO_3)_2, 6H_2O$)

4. Correct "ZIF8 NPs were produced succeeding in the recently revealed sol-gel method.²⁵" to "ZIF8 NPs were produced following the recently described sol-gel method [25]."
5. Correct "A serious difference in around 60% weight reduction after 500 °C recommends the separation of the natural ligand which accounts for collapse of the system structure." To "A serious difference in around 60% weight reduction after 500 °C suggests the separation of the natural ligand which accounts for the collapse of the system structure."
6. Correct "To investigate the optical properties of the frameworks" to "To discover the optical properties of the frameworks,"
7. Correct "To discover the optical properties of the frameworks, UV-Vis absorption bands were estimated for every sample." To "To investigate the optical properties of the frameworks, UV-Vis absorption bands were measured for every sample."
8. Correct "as appeared in Figure 4a (Blue)" to "as appears in Figure 4a (Blue)"
9. Correct "become a useful tool.[36, 37] The" to "become a useful tool [36, 37]. The"

Reviewer: 2

Comments to the Author(s)

The article entitled "Nano MOFs as targeted drug delivery agents to combat antibiotic resistant bacterial infections." has provided novel nanomaterials for the treatment of drug resistant bacteria. The authors suggest that the utilization of ZIF8 to load antibiotics RF can help with bacteria target delivery of antibiotics with enhanced antibacterial efficacy. Overall, this is an interesting study and it is within the journal's scope, however the evidence are not so sufficient and there are some serious issues to be addressed.

1. The authors claimed that the size of ZIF8 was unchanged before and after drug loading, however in the size of nanoparticles of ZIF8 and ZIF8-RF seemed much different shown in Figure 2 a and b, please explain.
2. The releasing profile of 8 h can only reach 0.2 even at pH 5.0, the authors should provide the drug releasing profile for longer periods.
3. Figure 3 please use the same magnification of ZIF8 and ZIF8-RF.
4. RF is not a common antibiotic used in clinic for treating MRSA, why not using regular antibiotics such as ampicillin? Moreover, the authors should test more than one drug to support the idea of this nano-MOF as target drug carrier.
5. What's the drug loading rate of ZIF8-RF? What's the concentration of RF used as control? What's the reason for the enhanced antibacterial efficacy of ZIF8-RF towards bacterial than pure RF? The authors should also provide the antibacterial effect of ZIF8 as control group and explain the enhanced antibacterial efficacy in detail.
6. The authors claimed that the nano-MOF can be used to combat drug resistant bacteria, however the antibacterial mechanism of this new ZIF8-RF seems to rely on RF, which is also an antibiotic. And ZIF8 only serves as a drug delivery system, how can this system overcome drug resistance?
7. What's the pH of the culture medium at the end of the treatment? As we know that the pH of LB culture medium is 7.4, it is difficult to believe that 3 h treatment in it this environment can trigger the release of RF in ZIF8-RF.
8. There is still much bacteria after the treatment of C3, as we know that longterm release of antibiotics at low concentrations can stimulate the emergence of drug-resistant bacteria, the authors should discuss this in the manuscript and provide the MIC for the ZIF8-RF.

9. In Figure 6 for the assay of biofilm, the authors didn't provide the anti-biofilm effect of ZIF8 and RF as control.

10. Spelling mistake, such as Page 4 line 10, "dangerto".

Author's Response to Decision Letter for (RSOS-200959.R0)

See Appendix A.

RSOS-200959.R1 (Revision)

Review form: Reviewer 1

Is the manuscript scientifically sound in its present form?

Yes

Are the interpretations and conclusions justified by the results?

Yes

Is the language acceptable?

Yes

Do you have any ethical concerns with this paper?

No

Have you any concerns about statistical analyses in this paper?

No

Recommendation?

Accept as is

Comments to the Author(s)

The authors have made the necessary changes to the manuscript and I think that it may be accepted in its current form.

Review form: Reviewer 2

Is the manuscript scientifically sound in its present form?

Yes

Are the interpretations and conclusions justified by the results?

Yes

Is the language acceptable?

Yes

Do you have any ethical concerns with this paper?

No

Have you any concerns about statistical analyses in this paper?

No

Recommendation?

Accept as is

Comments to the Author(s)

The author answered the questions clearly and revised the article.
I recommended the publication with no further changes.

Decision letter (RSOS-200959.R1)

Dear Professor Pal:

Title: Nano MOFs as targeted drug delivery agents to combat antibiotic resistant bacterial infections

Manuscript ID: RSOS-200959.R1

It is a pleasure to accept your manuscript in its current form for publication in Royal Society Open Science. The chemistry content of Royal Society Open Science is published in collaboration with the Royal Society of Chemistry.

RSC Associate Editor:
Comments to the Author:
(There are no comments.)

RSC Subject Editor:
Comments to the Author:
(There are no comments.)

Reviewer(s)' Comments to Author:
Reviewer: 1

Comments to the Author(s)
The authors have made the necessary changes to the manuscript and I think that it may be accepted in its current form.

Reviewer: 2

Comments to the Author(s)
The author answered the questions clearly and revised the article.
I recommended the publication with no further changes.

Appendix A

Response to the Reviewers

The authors would like to thank the reviewers for valuable comments. Indeed, the constructive comments by reviewers which we have addressed point by point turn out to be extremely useful in clarifying and improving the current manuscript.

Reviewer 1:

Comments:

*Professor Saleh and his coworkers report work in which they attach rifampicin (RF), a well-recognized antituberculosis drug, into the pore surface of zeolitic imidazolate framework 8 (ZIF8) by a simple and convenient synthetic procedure. The authors used X-ray diffraction (XRD) method to characterize the synthesized ZIF8 before and after drug encapsulation. In addition, they utilized scanning electron microscope (SEM) and transmission electron microscope (TEM) methods to characterize the binding between ZIF8 and RF. The authors have found that ZIF8-RF depicts pH-responsive drug delivery and is ideal for targeting bacterial disease due to its inherent acidic nature. One key finding, ZIF8-RF has been observed to display enhanced antibacterial activity against methicillin-resistant *S. aureus* (MRSA) bacteria and also inflicts full damage on structurally robust bacterial biofilms. The present study offers extensive physical insight for the manufactured antibiotic-encapsulated nanohybrids which are promising prospects and may prove beneficial for diverse practical applications.*

General comments: *I appreciate Scheme 1: Schematic representation of nano MOFs as targeted drug delivery agents to combat antibiotic-resistant bacterial infections. It is very informative. The research carried out by Dr. Saleh et al. will appeal to wide audience in areas such as medicinal chemistry, organic chemistry, drug design and drug delivery. The research is original and has not been published elsewhere. The present work contributes to the advancement of scientific knowledge. All conclusion drawn by the authors are supported by the data and data supporting the findings of the paper have been included in the manuscript. I believe that all experimental protocols/procedure and statistical analysis performed to a high technical standard which are both methodologically and scientifically sound. Characterisation of new compounds also meets the criteria outlined by the journal.*

The manuscript is well suited for The Royal Society Open Science.

Reply: We would like to thank the learned reviewer for appreciating our work. As per the kind concern of the learned reviewer, we corrected the required changes, and the same are represented in the revised manuscript.

Query 1: *Add spaces between text and references throughout the manuscript. For example, wellbeing[8] should be corrected to wellbeing [8].*

Reply: Authors would like to thank the learned reviewer for pointing out the issues. As per the kind suggestion of the learned reviewer, we have added spaces between text and references throughout the manuscript (Page No.3, Line No. 3, 7, 12, 15, 16, 19, 21 and 22; Page No. 4, Line No. 3, 4, 8, 11, 14, 17, 19, 21 and 23; Page No. 5, Line No. 2; Page No. 6, Line No. 2 and 16; Page No. 7, Line No. 2; Page No. 8, Line No. 7 and 11; Page No. 9, Line No. 2; Page No. 10, Line No. 15,16 and 18; Page No. 11, Line No. 5, 9 and Page No. 12, Line No. 5).

Query 2: *Change “We have also earlier showed that” to “We have showed earlier”*

Reply: The authors would like to thank the learned reviewer for this concern. As per the kind suggestions of the learned reviewer the issue has been addressed in the revised manuscript (Page No. 4, Line No. 17).

Query 3: *Remove spaces in the formula (Zn (NO3)2, 6H2O*

Reply: The authors highly appreciate the learned reviewer's concern. We have removed the spaces like $Zn(NO_3)_2, 6H_2O$ in the revised manuscript (Page No. 5, Line No. 17; Page No. 6, Line No. 3).

Query 4: *Correct “ZIF8 NPs were produced succeeding in the recently revealed sol–gel method.25” to “ZIF8 NPs were produced following the recently described sol–gel method [25].*

Reply: The authors are highly appreciated by the learned reviewer's concern. As per the kind suggestion of the learned reviewer the sentence has been corrected in the revised manuscript (Page No. 6, Line No. 2).

Query 5: *Correct “A serious difference in around 60% weight reduction after 500 °C recommends the separation of the natural ligand which accounts for collapse of the system structure.” To “A serious difference in around 60% weight reduction after 500 °C suggests the separation of the natural ligand which accounts for the collapse of the system structure.”*

Reply: The authors are highly appreciated the learned reviewer's concern. As per the kind suggestion of the learned reviewer the sentence has been corrected in the revised manuscript (Page No. 8, Line No. 14-16).

Query 6: Correct “To investigate the optical properties of the frameworks” to “To discover the optical properties of the frameworks,”

Reply: The authors are highly appreciated the learned reviewer's concern. As per the kind suggestion of the learned reviewer the sentence has been corrected in the revised manuscript (Page No. 10, Line No. 1-2).

Query 7: Correct “To discover the optical properties of the frameworks, UV-Vis absorption bands were estimated for every sample.” To “To investigate the optical properties of the frameworks, UV-Vis absorption bands were measured for every sample.”

Reply: The authors would like to thank the learned reviewer for this concern. As per the kind suggestions of the learned reviewer this issue has been addressed in the revised manuscript (Page No. 10, Line No. 1-2).

Query 8: Correct “as appeared in Figure 4a (Blue)” to “as appears in Figure 4a (Blue)”

Reply: The authors would like to thank the learned reviewer for this concern. As per the kind suggestions of the learned reviewers this issue has been modified in the revised manuscript (Page No. 10, Line No. 4-5).

Query 9: Correct “become a useful tool.[36, 37] The” to “become a useful tool [36, 37]. The”

Reply: The authors are highly appreciated by the learned reviewer's concern. As per the kind suggestion of the learned reviewer the sentence has been corrected in the revised manuscript (Page No. 10, Line No. 18).

Reviewer 2:

Comments:

The article entitled “Nano MOFs as targeted drug delivery agents to combat antibiotic resistant bacterial infections.” has provided novel nanomaterials for the treatment of drug resistant bacteria. The authors suggest that the utilization of ZIF8 to load antibiotics RF can help with bacteria target delivery of antibiotics with enhanced antibacterial efficacy. Overall, this is an interesting study and it is within the journal’s scope, however the evidence are not so sufficient and there are some serious issues to be addressed.

Reply: The authors would like to thank the learned reviewer for appreciating our work. As per the kind concern of learned reviewer, we addressed the required points and the same are represented in the revised manuscript.

Query 1: *The authors claimed that the size of ZIF8 was unchanged before and after drug loading, however in the size of nanoparticles of ZIF8 and ZIF8-RF seemed much different shown in Figure 2 a and b, please explain.*

Reply: The authors would like to thank the learned reviewer. SEM images of ZIF8 and ZIF8-RF are represented using different magnification. So, only observing the images would imply different sizes. However, performing the size distribution profile in both the images ultimately results in a similar size of nano-systems as shown in Figure R1.

Figure R1: (a) The FESEM image of ZIF8. (b) FESEM image of ZIF8-RF. (c) Particle distributions of ZIF8 (average particle size 158.72 ± 0.52 nm). (d) Particle distributions of ZIF8-RF (average particle size 157.96 ± 1.07 nm).

Figure R1(c) represents the particle size distribution of ZIF8 (average particle size 158.72 ± 0.52 nm) and Figure R1(d) represents the particle size distribution of ZIF8-RF (average particle size 157.96 ± 1.07 nm). The size of ZIF8 was unchanged before and after drug loading. The issue has been addressed in the revised manuscript (Page No. 9, Line No. 9-12; Page No. 19, Figure 2).

Query 2: The releasing profile of 8 h can only reach 0.2 even at pH 5.0, the authors should provide the drug releasing profile for longer periods.

Reply: The authors would like to thank the learned reviewer. We appreciate the reviewer's concern. Although the drug releasing profile of 8 h can reach 0.2 at pH 5.0, the RF releasing percentage is 69% with respect to control ZIF8, which is significant. It has to be noted that a drug which is not being released for a longer time (between 5 and 12 h after the administration of single dose) may be excreted from our body (Chemotherapy 16.6, 1971, 356-370.). The above reason is not inspired to perform longer drug releasing experiment. Moreover, in the current pandemic (COVID19) situation our lab is closed and we will not be able to perform experiments. However, the kind suggestion of the learned reviewer will be included in our future studies. The issue has been addressed in the revised manuscript (Page No. 11, Line No. 1-4).

Query 3: Figure 3 please use the same magnification of ZIF8 and ZIF8-RF.

Reply: The authors would like to thank the learned reviewer for this concern. We have performed the EFTEM mapping in ZIF8 and ZIF8-RF samples. We have determined the elemental composition from the images. There is no motive to compare between the two structures. Thus, we took the mapping in suitable signal conditions.

Query 4: RF is not a common antibiotic used in clinic for treating MRSA, why not using regular antibiotics such as ampicillin? Moreover, the authors should test more than one drug to support the idea of this nano-MOF as target drug carrier.

Reply: The authors would like to thank the learned reviewer for this concern. Rifampicin is a widely used antibiotic for the treatment of several types of bacterial infections, including tuberculosis, Mycobacterium avium complex, leprosy, and Legionnaires' disease. We used Rifampicin for this issue. We appreciate reviewer's concern to test more than one drug to support the idea of targeted drug carrier but few earlier literatures are comprising of different types of drug encapsulated ZIF-8 showing pH sensitive drug release (ACS Appl. Bio Mater. 2019, 2, 4, 1772–1780, Biomed. Phys. Eng. Express. 2018, 4, 055004). So, we chose only Rifampicin and evaluate the antibacterial efficacy of the system.

Query 5: *What's the drug loading rate of ZIF8-RF? What's the concentration of RF used as control? What's the reason for the enhanced antibacterial efficacy of ZIF8-RF towards bacterial than pure RF? The authors should also provide the antibacterial effect of ZIF8 as control group and explain the enhanced antibacterial efficacy in detail.*

Reply: The authors would like to thank the learned reviewer for this concern. The RF drug loading capacity (DLC) was measured in an ethanol solution of RF and by comparing the absorption peak intensity of RF at 480 nm before and after loading onto ZIF8 sample. The DLC (%) was calculated according to the equations below:

$$\frac{\text{amount of drug loaded}}{\text{amount of ZIF8 sample}} \times 100$$

The DLC of ZIF8 was found to be 42%. Such high DLC is crucial for therapeutic applications. The issue has been addressed in the revised manuscript (Page No. 6, Line No. 9-13 and Page No. 11, Line No. 11-12).

The equal concentrations of RF (maintaining the O.D. at 480 nm similar to the ZIF8-RF) were used for determinations of the effect of RF in antibacterial effect.

We have also performed the ZIF-8 control group which shows no significant antibacterial effect. Moreover, It has also been reported in our previous article (Biomedical Physics & Engineering Express 4.5 (2018): 055004) that ZIF8 (low concentration) showed minimal antibacterial effect compared to drug and drug loaded MOF. Thus, we have not included the data into the figure and compared the efficacy of RF with the encapsulated systems.

The improved efficacy of ZIF8-RF is due to the greater availability of RF into bacterial infection sites having lower pH. The better bioavailability of RF at the infection site in the case of ZIF8-RF improves its action compare to free RF.

Query 6: *The authors claimed that the nano-MOF can be used to combat drug resistant bacteria, however the antibacterial mechanism of this new ZIF8-RF seems to rely on RF, which is also an antibiotic. And ZIF8 only serves as a drug delivery system, how can this system overcome drug resistance?*

Reply: The authors would like to thank the learned reviewer's concern. We have tested the antibacterial effects using MRSA strain and the system shows good results. We are not sure how it will work on different antibiotic resistant strains. We believe a wide-spread study is required before commenting on its overall effect on drug-resistance issues.

Query 7: What's the pH of the culture medium at the end of the treatment? As we know that the pH of LB culture medium is 7.4, it is difficult to believe that 3 h treatment in it this environment can trigger the release of RF in ZIF8-RF.

Reply: The authors would like to thank the learned reviewer for the concern. It has to be noted that pH of LB culture medium reduces with bacterial growth (PLoS biology 16.3, 2018). We have performed control experiments using same amount of ZIF-8 to that of ZIF8-RF with 3 hr incubation time. ZIF-8 has not provided any significant change in number of bacterial colonies compared to the untreated MRSA. This effect suggests that delivery of RF plays a significant role in the antibacterial effect of ZIF8-RF composite. Thus, we anticipated the lower pH condition due to bacterial growth, promotes the loosening of bonds in porous ZIF8 and consequently improve the delivery of Rifampicin. We have addressed the issue in the revised manuscript (Page No. 11, Line No. 13-20).

Query 8: There is still much bacteria after the treatment of C3, as we know that longterm release of antibiotics at low concentrations can stimulate the emergence of drug-resistant bacteria, the authors should discuss this in the manuscript and provide the MIC for the ZIF8-RF.

Reply: The authors would like to thank the reviewer for the concern. The RF loaded MOF has enhanced antibacterial activity. To get the appropriate or proper antibacterial activity of ZIF8-RF we have performed triplicate of experiments for CFU assay and plotted the CFU bar with proper standard deviations. The MIC was found to be 1 mg/mL for ZIF8-RF samples as shown in Figure R2.

Figure R2: Dose-dependent antibacterial effect of ZIF8-RF at varying concentrations on methicillin-resistant *S. aureus*.

We represent here a proof of concept about using ZIF8 MOF as delivery agent for antibiotics and its subsequent use for antibacterial and antibiofilm effects. We will expand the scope of the present study using a greater number of antibiotics, range of drug concentration and more bacterial strains to comment on its overall effects in the near future. This issue has been addressed in the revised manuscript (Page No. 12, Line No. 1; Page No. 23, Figure No. 6b).

Query 9: In Figure 6 for the assay of biofilm, the authors didn't provide the anti-biofilm effect of ZIF8 and RF as control.

Reply: The authors would like to thank the learned reviewer's concern. As the antibacterial effect of ZIF8-RF conjugates is severe than ZIF8 or rifampicin alone, so we have proceeded with ZIF8-RF to check the antibiofilm effects.

Query 10: Spelling mistake, such as Page 4 line 10, "dangerto".

Reply: The authors would like to thank the reviewer for the concern. We have corrected the spelling "dangerto" to "danger to" and the same has been incorporated in the revised manuscript (Page No. 3, Line No. 4).